# Study of a Bimodal α–β Ti Alloy Microstructure Using Multi-Resolution Spherical Indentation Stress-Strain Protocols

**Natalia Millan-Espitia** [1] and **Surya R. Kalidindi** [1,2,*]

1    George W. Woodruff School of Mechanical Engineering, Georgia Institute of Technology,
     Atlanta, GA 30332, USA; natalia.millan@me.gatech.edu
2    School of Computational Science and Engineering, Georgia Institute of Technology, Atlanta, GA 30332, USA
*    Correspondence: surya.kalidindi@me.gatech.edu

**Abstract:** Recent investigations have highlighted the multi-resolution and high throughput characteristics of the spherical indentation experimental and analysis protocols. In the present work, we further demonstrate the capabilities of these protocols for reliably extracting indentation stress-strain (ISS) responses from the microscale constituents as well as the bulk scale of dual phase materials exhibiting bimodal microstructures. Specifically, we focus on bimodal microstructures produced in an α–β Ti6242 sample. Combining the multi-resolution indentation responses with microstructural statistics gathered from the segmentation of back-scattered electron images from the scanning electron microscope allowed for a critical experimental evaluation of the commonly utilized Rule of Mixtures based composite model for the elastic stiffness and plastic yield strength of the sample. The indentation and image analyses protocols described in this paper offer novel research avenues for the systematic development and critical experimental validation of composite material models.

**Keywords:** composite material; bimodal microstructure; titanium; Ti6242; spherical indentation; image segmentation; rule of mixtures; effective property

## 1. Introduction

Microstructures exhibiting two distinct morphologies in the arrangement of their phase constituents are generally referred to as bimodal microstructures [1,2]. Most commonly, these microstructures exhibit single-phase equiaxed grains (refers to regions of uniform crystal lattice orientation in the material microstructure) alongside grains displaying distinct dual-phase morphologies (e.g., lamellar, dendritic). Examples of such microstructures can be seen in two-phase steels, alpha-beta brasses, alpha-beta titanium alloys, and bulk metallic glass-matrix composites. This class of composite microstructures offers tremendous potential for several advanced technology applications since their effective properties can be tailored more easily to meet the designer-specified targets through the modulation of the relative volume fractions and morphological features of their phase constituents.

Our interest in this paper will be on the bimodal microstructures in α–β Ti alloys. It is well known that the microstructures in these alloys can be controlled through suitably designed heat treatments. Generally, one can take the material above the beta-transus temperature (i.e., temperature where the hcp α transforms completely into bcc β) and cool the material down at different cooling rates to produce a variety of distinct microstructures. One can also apply heat treatments at temperatures below the beta-transus temperature to allow for the formation and stabilization of fully α-phase regions known as primary α grains. Slow cooling generally leads to the formation of parallel secondary α-laths (i.e., colonies), whereas higher cooling rates favor the formation of the crisscrossed secondary α-laths (i.e., basket-weave morphology) [3,4]. It is therefore possible to produce a broad range of bimodal microstructures in α–β Ti alloys through tailored heat treatments. Bimodal microstructures in α–β Ti alloys have been reported to result in improved mechanical

properties. As a specific example, a bimodal microstructure consisting of approximately 30 vol.% of equiaxed primary α grains and 70 vol.% of lamellar α–β has been found to provide an optimal combination of creep and fatigue properties for compressor disks operating at high temperatures [3]. Besides the influence of phase morphologies, the mechanical response of α–β Ti alloys is also strongly influenced by the grain-scale anisotropy of the hcp α, which is known to exhibit a pronounced crystal lattice orientation dependence [5–10].

Bimodal microstructures have been studied extensively in prior literature in efforts to understand and predict their effective properties [11–23]. Towards this goal, it is important to accurately measure the mechanical properties at different length scales of the material. Of primary interest are the effective properties at the scale of a representative volume element (RVE) of the material microstructure (i.e., length scales covering multiple grains and colonies) and properties at the scale of individual constituents (i.e., individual grains or colonies). At the smaller length scales, protocols involving micropillar compression [24–28], micro-cantilever beam bending [29–31], and micro-hardness tests [28,32–35] have been shown to provide valuable mechanical response information for the individual constituents in these materials. However, these protocols often incur high costs, require tedious sample preparation procedures, and produce limited amount of data [5,36]. Moreover, hardness tests produce hardness values that are not easily related to other standard mechanical properties of the constituents. This is due to the fact that the measured values often exhibit large variances in the results produced by different research groups due to the inherent variances in the test and analyses protocols employed [5,37–41]. At the larger length scales (i.e., RVE scale), the effective properties are typically measured using standard protocols such as uniaxial tension [42] and compression [43]. The lack of consistency in the experimental protocols employed at the different material length scales is likely to have contributed significantly to the large discrepancies in the multiresolution mechanical properties reported for several advanced material systems [13,14,34,35,38,39,44–51]. As a specific example, the micro-hardness measurements of Gupta et al. [48] on basket-weave grains in Ti64 specimens suggested uniaxial yield strength values in the range 950–1308 MPa [39], which are significantly higher than the macroscale tensile yield strengths measured in the range of 905–945 MPa by Sieniawski et al. [49] on similar samples (i.e., samples with similar processing histories). These discrepancies could arise from the differences in the loading histories (i.e., indentation versus tension) in the measurement protocols employed at the different material structure scales.

In efforts aimed at establishing consistent and reliable multi-resolution mechanical testing capabilities, Kalidindi and Pathak [52–54] have developed the spherical indentation protocols based on the Hertzian theory of elastic contact between two deformable isotropic solids with quadratic surfaces [55]. These protocols capture the mechanical responses in the sample at different length scales in the form of load-displacement curves, and subsequently convert them to indentation stress-strain curves (ISS). At the grain-scale, these protocols were successfully demonstrated to capture the dependence of the local mechanical response on the grain orientation in different material classes [5,7,56–60]. At the macroscale, a consistent set of measurement and analyses protocols were successfully demonstrated for the reliable evaluation of the effective mechanical properties of the material [37,61–66]. An attractive feature of these protocols is that they incur significantly less effort and cost compared to many of the other competing protocols mentioned above, while requiring relatively small material volumes.

Consistent multi-resolution mechanical test protocols are central to the systematic evaluation and refinement of composite theories capable of predicting the effective properties of a material based on the details of the material microstructure and the individual properties of the microscale constituents. As a specific example, the simple rule of mixtures (ROM) model is commonly employed [13,16–23], but has only been critically evaluated in only a few experimental studies [13,16,20–23]. As already discussed earlier, the main hurdle comes from the lack of consistent protocols that can be applied at the different material structure length scales. In recent work [20–22], the ROM model was confirmed to

exhibit impressive accuracy in estimating the effective yield strength of ferrite-martensite steels [20–22], in spite of the highly simplified assumptions implied by this model.

In this work, we aimed to critically evaluate the performance of the ROM model for estimating the effective yield strength of a bimodal Ti6242 specimen. This was accomplished using multi-resolution spherical nanoindentation protocols on a bimodal sample exhibiting primary $\alpha$ and basket-weave $\alpha$–$\beta$ morphologies. Multi-resolution microstructure characterization of the samples was conducted using scanning electron microscopy and electron backscatter diffraction. It is shown that the consistent use of the multiresolution spherical indentation protocols on the bimodal Ti6262 microstructure produces grain-scale and macroscale measurements of the yield strengths in the sample that are fairly consistent with the ROM model.

## 2. Materials and Methods

### 2.1. Materials and Sample Preparation

The material chosen for this study was Ti-6Al-2Sn-4Zr-2Mo (Ti6242) due to its versatility in producing diverse microstructures through suitable heat treatments. The actual alloy composition was determined in a previous study by Pilchak et al. [67] to be Ti-5.93Al-2.01Sn-4.05Zr-1.88Mo-0.12Si, with the interstitial contents of oxygen, iron, and nitrogen being 0.107, 0.05, and 0.001 in wt.%, respectively. Atomic absorption and inductively coupled plasma mass spectrometry was employed to determine the metallic element composition, while inert gas fusion was employed for the measurement of the smaller elements. Specimens of dimensions 10.0 mm $\times$ 20.0 mm $\times$ 2.0 mm were cut using an electric discharge machining and placed into quartz tubes, which were subsequently backfilled with Argon to protect the samples from oxidation during the heat treatment process. Heat treatment was designed such that the microstructure was composed of primary-$\alpha$ and basket-weave morphologies with large enough grains that allowed the application of the spherical indentation protocols. Specifically, the specimen was heat-treated at 986 °C (10 °C below the beta transus for a bimodal microstructure) for 6 h, followed by a water-quenching step to achieve the desired bimodal microstructures comprising the basket-weave components. Subsequently, the sample was subjected to a stress-relief heat treatment (700 °C for 4 h) and air-cooled. The heat treatment was conducted in a Thermo Fisher Scientific, Lindberg/Blue 1100 °C Box furnace.

Specimen surfaces were prepared for microscopy and indentation following standard metallography protocols procedures [68]. These protocols removed any oxide or mechanically deformed layers. Specifically, for this work, it was important to minimize the height disparities on the sample surface due to the unavoidable differences in the polishing rates of the different microscale constituents present in the sample. Chemo-mechanical preparation steps utilized on these specimens included surface grinding using silicon carbide papers (starting with 800 grit and systematically going to 2400 grit), polishing using 9 $\mu$m, 3 $\mu$m and 1 $\mu$m diamond suspension on a Struers' (Copenhagen, Denmark) Tegramin Automatic Grinding Machine, and final polishing on the Buehler's (Lake Bluff, IL, USA) Vibremet 2 Vibratory polisher for 12 h with a medium consisting of one part of 0.06 $\mu$m colloidal silica, 4 parts of water, and 1 part of hydrogen peroxide.

### 2.2. Spherical Indentation

Mechanical characterization of the samples was conducted at room temperature on a Keysight Nano-Indenter G200 with the Continuous Stiffness Measurement (CSM) option. Two different tip sizes were used in this study: (i) a 15.2 $\mu$m radius tip was employed for the grain scale characterization of primary $\alpha$ and basket-weave (prior $\beta$) grains, and (ii) a 500 $\mu$m radius tip was used for the characterization of the bulk response of the sample over regions comprising a large number of grains. The grain-scale indentations were placed close to the centers of the grains, and only one indentation was conducted per grain to avoid proximity with the grain boundary and other previous indentations in the sample. For the indentations on the primary $\alpha$ grains, the average contact radius at yield

(over grains of different orientations) was estimated to be about 550 nm (using equations presented in the next subsection). Since this is significantly smaller than the average size of the primary α grains of about 10 μm, these measurements are assumed to reflect the local grain-scale mechanical responses. For the indentations on the basket-weave grains, the average contact radius at yield was estimated to be about 640 nm. The deformation zone size in these indentations is significantly smaller than the prior β grain size of about 16 μm, but larger than the average α lath thickness of about 18 nm (lath spacing is significantly smaller in these microstructures). Therefore, the measurements in the basket-weave grains are assumed to reflect the effective properties of the basket-weave components. For the indentations with the larger tip, the average contact radius at yield was estimated to be about 25 μm. The deformation zone in these indentations are expected to include about 13 grains (mixtures of primary α and basket-weave components), and therefore an ensemble average of these measurements is assumed to reflect the bulk response of the sample. A total of 150 indentation tests were conducted for this work (about 50 tests on each grain scale morphology and about 50 tests for the bulk response).

### 2.3. Spherical Nanoindentation Analysis

The spherical indentation experimental and analysis protocols used in this work are largely based on Hertz theory, which is mainly focused on describing the deformation during frictionless contact between two approaching elastic bodies with quadratic surfaces [55]. The load-displacement relationship for such indentation is expressed as

$$P = \frac{4}{3} E_{eff} R_{eff}^{\frac{1}{2}} h_e^{\frac{3}{2}} \tag{1}$$

$$\frac{1}{E_{eff}} = \frac{1 - v_i^2}{E_i} + \frac{1 - v_s^2}{E_s} \tag{2}$$

$$\frac{1}{R_{eff}} = \frac{1}{R_i} + \frac{1}{R_s} \tag{3}$$

where $P$, $E_{eff}$, $R_{eff}$, and $h_e$ denote the indentation load, effective elastic modulus, effective radius, and elastic indentation displacement, respectively (see Figure 1a).

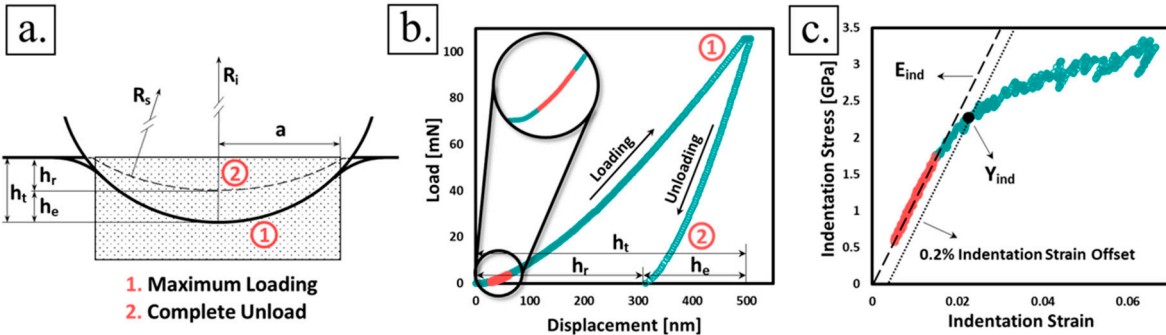

**Figure 1.** (**a**) Schematic description of the spherical indentation experiment depicting the loaded and unloaded configurations. (**b**) Measured load-displacement curve from a typical spherical indentation measurement, with a zoomed view of the early loading segment. (**c**) Indentation stress-strain (ISS) curve extracted from the load-displacement data with the elastic region of the deformation highlighted in red.

Furthermore, $R$, $E$, and $v$ denote the radius, Young's modulus and the Poisson's ratio, respectively, while the subscripts $i$ and $s$ correspond to the indenter and sample, respectively. The value of $E_{eff}$ is estimated from a segment selected in the initial elastic loading [53] (the bright colored segment in Figure 1b), and is assumed to be unchanged throughout

the entire deformation of the material. In the analyses of the initial elastic segment for the determination of $E_{eff}$, $R_s$ is assumed to represent infinity (i.e., $R_{eff} = R_i$) [53].

The analysis of the indentation load-displacement data starts with a zero-point correction, which identifies an effective point of initial contact between the indenter and the sample by finding a segment of the load-displacement curve that complies with Hertz's theory (i.e., Equation (1)). This correction accounts for the unavoidable surface artifacts such as roughness and oxide layers that affect the indentation measurements [53]. The zero-point correction is implemented using regression techniques that fit the measurements to the following equation derived from Hertz's theory [53].

$$S = \frac{3P}{2h_e} = \frac{3\left(\widetilde{P} - P^*\right)}{2\left(\widetilde{h} - h^*\right)} \tag{4}$$

where $(\widetilde{P}, \widetilde{h})$ denote the measured load-displacement values, $(P^*, h^*)$ denote the zero-point load and displacement corrections, respectively, and $S$ denotes the CSM (obtained here using a superimposed load-unloading cycles of 2 nm amplitude and 45 Hz frequency) [69,70].

The use of CSM allows for the estimation of the contact radius, $a$, along the complete monotonic loading history, which can then be used to estimate the indentation stress and indentation strain values corresponding to every point on the measured load-displacement curve (see Figure 1b,c). These computations are performed using

$$a = \frac{S}{2E_{eff}} \tag{5}$$

$$\sigma_{ind} = \frac{P}{\pi a^2} \tag{6}$$

$$\varepsilon_{ind} = \frac{4}{3\pi} \frac{h_t}{a} \tag{7}$$

The indentation yield strength, $Y_{ind}$, is defined as the indentation stress at a 0.2% offset indentation plastic strain in the indentation stress-strain (ISS) curve (see Figure 1c).

### 2.4. Microstructure Characterization and Quantification

The back-scattered electron imaging signal from a Tescan Mira3 field emission scanning electron microscope (FE-SEM) was used for the characterization of the bimodal microstructure in the sample. Images with a constant view-field of 100 μm and size $2048 \times 2048$ pixels were captured using an accelerating voltage of 15 kV. The acquired grayscale images were analyzed through a series of image processing steps in order to label each pixel in the image as belonging to one of the two grain-scale morphologies (i.e., primary $\alpha$ or basket-weave). This process is generally referred to as image segmentation, and the sequence of image processing steps used in this process are referred as segmentation workflows [71–74]. The segmentation workflows utilized in this work were based on the framework proposed recently by Iskakov and Kalidindi [75]. They comprised the following steps: (i) gaussian global noise removal using the *imgaussfilt* function [76] with a smoothing kernel whose standard deviation was set to 0.9; (ii) global thresholding step with the *imquantize* function for which a single specified quantization value of 85 was selected [76]; and (iii) post-processing steps such as the *bwareaopen* function [76] to remove all connected objects that have fewer than 1000 pixels and the *imclose* function [76,77] which performs a dilation-erosion dual operation on the segmented areas using a disk of 3 pixel radius. Segmentation validation involved visual validation, as well as a more quantitative approach using precision and recall scores [78]. The quantitative validation of the segmented image was used to determine the parameter values mentioned above. An example of the application of our segmentation workflow is presented in Figure 2.

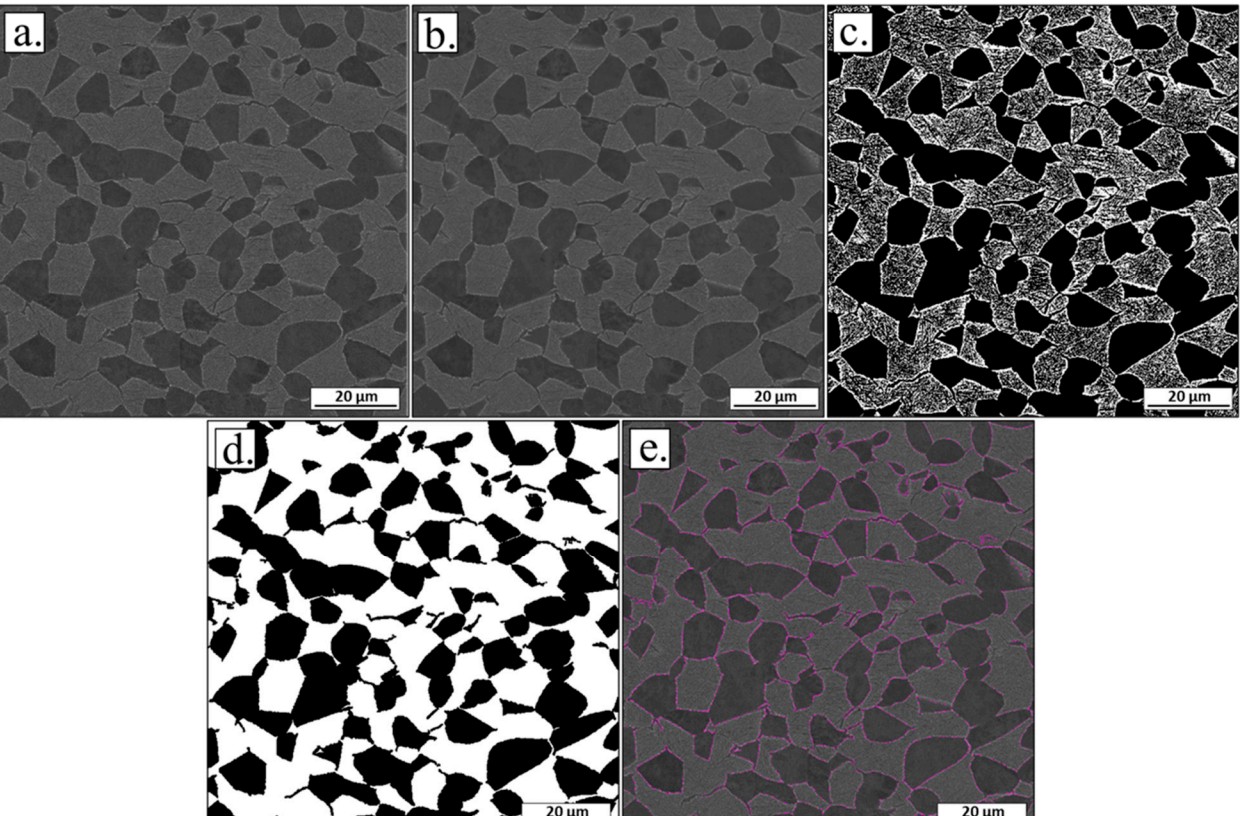

**Figure 2.** Segmentation workflow applied on a BSE micrograph with view-field of 100 μm. (**a**) Raw image from the scanning electron microscope device. (**b**) Image after a gaussian global noise removal correction. (**c**). Resulting micrograph after the thresholding step. (**d**) Final segmented microstructure representative. (**e**) Original image with the phase boundaries highlighted in magenta for the visual validation of the segmentation process.

The crystallographic orientations of the primary $\alpha$ phase grains were measured using electron back-scattered diffraction (EBSD) mapping in the SEM using an accelerating voltage of 20 kV. A total of 20 EBSD scans of adjacent and slightly overlapping areas of 150 μm × 150 μm were imaged using a 0.5 μm step-size. Using the TSL OIM Analysis 8 software, the images were suitably stitched to create a single large scan of approximately 680 μm × 545 μm (see Figure 3a). In order to extract the texture information for the primary $\alpha$ phase, the image quality parameter was used to filter out the pixels corresponding to the basket-weave grains as they exhibit low image quality values. The EBSD map showing the orientations of the primary $\alpha$ grains is shown in Figure 3b. It is seen that the primary $\alpha$ regions in the sample exhibit a pronounced texture. The area fractions corresponding to 10-degree bins in the declination angle ($\Phi$) were computed and are summarized in Table 1.

**Table 1.** Fraction corresponding to each of the binned regions of the primary a phase, and the corresponding volume fractions of all constituents in the bulk microstructure.

| Bimodal Microstructure Statistics | | |
|---|---|---|
| **Local State** | **Fraction in $\alpha$-Phase** | **Fraction in Bulk ($f$)** |
| $\alpha = 0°–10°$ | 16% | 6.8% |
| $\alpha = 10°–20°$ | 6.1% | 2.6% |
| $\alpha = 20°–30°$ | 1.6% | 0.7% |
| $\alpha = 30°–40°$ | 2.0% | 0.8% |
| $\alpha = 40°–50°$ | 3.0% | 1.3% |
| $\alpha = 50°–60°$ | 9.7% | 4.1% |

**Table 1.** *Cont.*

| Bimodal Microstructure Statistics | | |
|---|---|---|
| Local State | Fraction in α-Phase | Fraction in Bulk (*f*) |
| α = 60°–70° | 17.7% | 7.5% |
| α = 70°–80° | 14.7% | 6.2% |
| α = 80°–90° | 29.1% | 12.3% |
| Basket-weave | 0% | 57.7% |

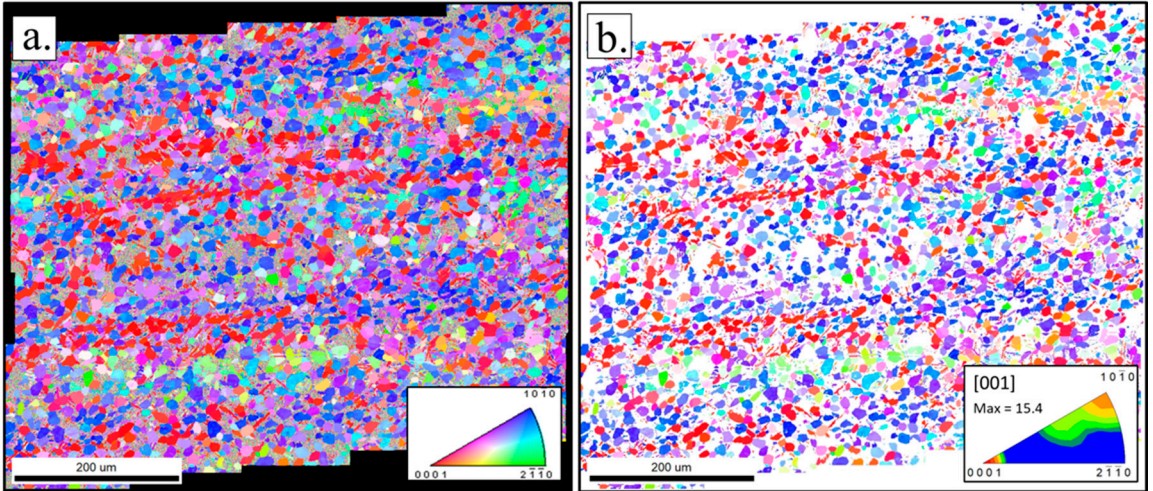

**Figure 3.** (**a**) Compilation of 20 EBSD scans from a Ti6242 bimodal microstructure with primary α and basket-weave grains. Areas with low image quality correspond to basket-weave grains. (**b**) Inverse Pole Figure map of the primary a grains only, with a harmonic tex.

## 2.5. Prediction of Effective Property

There exist several models in literature for the prediction of the effective elastic and plastic properties of composites based on microstructure statistics and the properties of their individual constituents. The classical Rule of Mixtures (ROM) is the simplest among these. For the bimodal titanium microstructure studied here, this model can be expressed as

$$P_{eff} = f_\alpha P_\alpha + f_{BW} P_{BW} \qquad (8)$$

where *P* and *f* refer to the property and volume fractions, respectively, subscripts α and *BW* refer to the constituent morphologies (primary a and basket-weave, respectively), and $P_{eff}$ denotes the effective property of the system.

This model has been evaluated extensively in literature [13,17–23] for predicting the effective yield strength of the composite. Due to the high plastic anisotropy associated with the primary α component of titanium alloys [5] (the yield strength of the basket-weave component does not exhibit a strong dependence on the lattice orientations of the α laths), in the present study we have re-written Equation (8) as

$$P_{eff} = f_{\alpha_{0°-10°}} P_{\alpha_{0°-10°}} + f_{\alpha_{10°-20°}} P_{\alpha_{10°-20°}} + \cdots + f_{\alpha_{80°-90°}} P_{\alpha_{80°-90°}} + f_{BW} P_{BW} \qquad (9)$$

where the different $f_\alpha$ have already been tabulated in Table 1. The corresponding $P_\alpha$ and $P_{BW}$ will be measured using the indentation protocols described above.

It is important to note that $P_{eff}$ is also measured using the same indentation protocols in our study. Therefore, the consistent high-throughput multi-resolution spherical indentation stress-strain protocols combined with microscopy protocols offer a unique opportunity to critically evaluate composite models.

## 3. Results and Discussion

### 3.1. Microstructure Statistics

From an ensemble of 10 segmented BSE-SEM images, the average primary $\alpha$ volume fraction in the sample was determined to be $42.3 \pm 4.1\%$. The remaining volume corresponding to the basket-weave morphology was computed as $f_{BW} = 1 - f_\alpha$. The inverse pole figure in Figure 3 and the values from Table 1 indicate that the primary $\alpha$ grain normals in our sample are predominantly aligned with the $[10\bar{1}0]$ and $[0001]$ directions in the hcp crystal. It is also clear that the primary $\alpha$ grains are arranged in a band-like structure, which is presumed to result from the prior thermo-mechanical deformation applied on the sample.

### 3.2. Spherical Nanoindentation Stress-Strain Measurements

The indentation stress-strain curves obtained from the application of the spherical indentation stress-strain protocols described above on the primary $\alpha$ grains, basket-weave grains, and the bulk measurements are presented in Figure 4. The values of the measured indentation moduli and indentation yield strengths in the primary $\alpha$ grains are correlated to their declination angles in Figure 5. In prior work, it has been shown that the influence of the other two Euler angles on these measurements is fairly low [5,7]. The results shown in Figure 5 are in good agreement with the values reported in a previous study, where similar protocols were applied on the primary $\alpha$ grains in a set of $\alpha$-, near-$\alpha$ and $\alpha$–$\beta$ titanium alloys [5]. The highest measured indentation yield strength in the primary $\alpha$ grains in our sample was observed to be about 3 GPa, and corresponded to the grains with their c-axis oriented parallel to the indentation direction. On the other hand, the lowest indentation yield strength was measured to be about 1 GPa for grains with the c-axis perpendicular to the indentation direction. Following the strategy outlined earlier, these measurements were binned by the declination angle and added to Table 2. Indentation modulus and indentation yield strength for each bin were determined by applying a polynomial regression using generalized spherical harmonics (GSH) on the complete dataset [79], and then establishing the values at the center-point of each bin, as depicted in Figure 5.

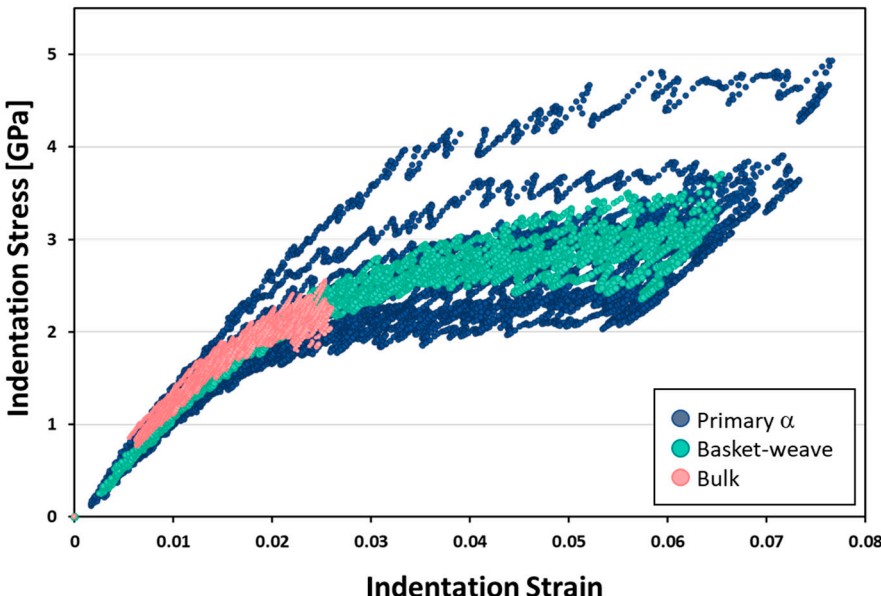

**Figure 4.** Examples of the indentation stress-strain curves measured in this study. The blue curves are from measurements on primary a grains, the green curves are from basket-weave grains, and the pink curves are bulk measurements.

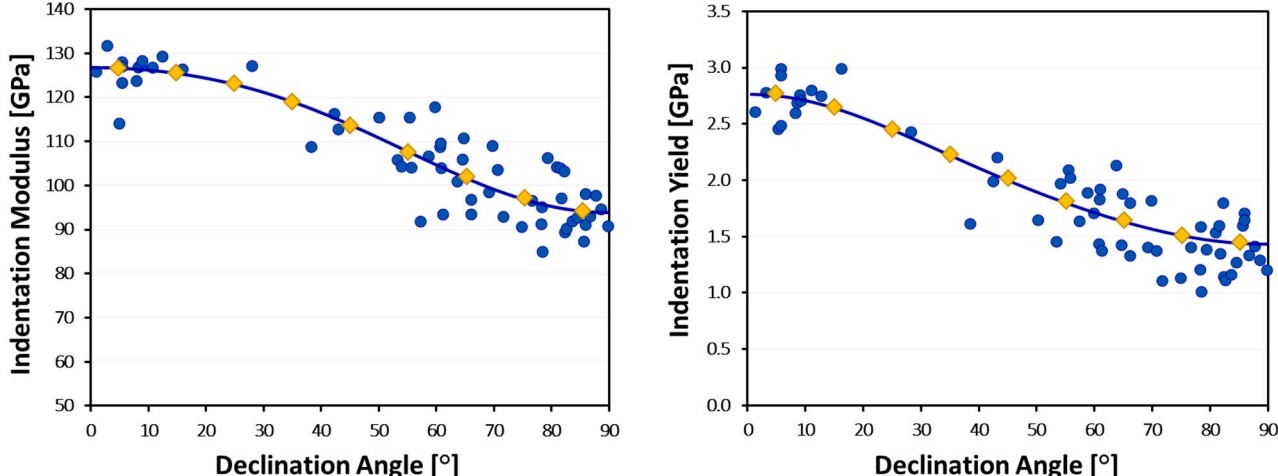

**Figure 5.** Results from the spherical indentations performed on the primary a grains, plotted as a function of the declination angle (Φ). Experimental results are shown as blue circles, and the values estimated from regression analysis at the mid points of each bin are depicted as yellow diamonds.

**Table 2.** Measured indentation moduli and indentation yield strengths for the different microscale constituents as well as the bulk responses of the material studied in this work. The last row provides the corresponding predictions from the application of the ROM model.

| Spherical Indentation Measurements | | | |
|---|---|---|---|
| **Length Scale** | **Morphology** | $E_{ind}$ **[GPa]** | $Y_{ind}$ **[GPa]** |
| | **Basket-Weave** | $121 \pm 3.1$ | $1.99 \pm 0.12$ |
| | $\Phi\alpha = 0°–10°$ | 125.2 | 2.73 |
| | $\Phi\alpha = 10°–20°$ | 124.1 | 2.65 |
| | $\Phi\alpha = 20°–30°$ | 121.7 | 2.47 |
| | $\Phi\alpha = 30°–40°$ | 118.4 | 2.25 |
| **Constituents** | $\Phi\alpha = 40°–50°$ | 111.9 | 2.03 |
| | $\Phi\alpha = 50°–60°$ | 106.6 | 1.79 |
| | $\Phi\alpha = 60°–70°$ | 100.9 | 1.60 |
| | $\Phi\alpha = 70°–80°$ | 96.3 | 1.44 |
| | $\Phi\alpha = 80°–90°$ | 93.7 | 1.37 |
| **Bulk** | **Bimodal** | $118 \pm 2.6$ | $1.96 \pm 0.10$ |
| ROM Predictions | | | |
| **Bulk** | **Bimodal** | 114 | 1.92 |

The average indentation modulus and the indentation yield strength for the basket-weave grains morphology were determined to be 121 GPa and 1.99 GPa, respectively (see Table 2). The corresponding standard deviations were computed as 3.12 GPa and 0.12 GPa, respectively. The measurements indicated a much-reduced variation in the mechanical properties exhibited by the basket-weave grains, when compared to the corresponding measurements on the primary $\alpha$ grains presented earlier. This observation suggests that the multiple $\alpha$ variants that coexist within a single basket-weave grain tend to homogenize the elastic and plastic properties to isotropic values at the grain-scale for this complex morphology.

The results from the application of the spherical indentation stress-strain protocols for the effective mechanical response are also summarized in Table 2. The average effective indentation modulus and the averaged effective indentation yield strength were measured to be 118 GPa and 1.96 GPa, respectively. The low standard deviations of these measurements (see Table 2) confirm that they reliably reflect the bulk response of the sample. Assuming an overall isotropic material response and using previously established

conversion rules [37,52,63,65,80], the sample's Young's modulus and the uniaxial yield strength are estimated as 106 GPa and 996 MPa, respectively. These are in good agreement with prior reports in literature using conventional uniaxial test methods. For example, Bertrand et al. [81] reported an uniaxial yield strength of 948 MPa for a similar Ti6242 bimodal sample.

*3.3. Evaluation of the Composite Model*

The volume fraction information presented in Table 1 along with the mechanical properties measured by spherical indentation presented in Table 2 were used for the evaluation of the composite model described in Equation (9). The predictions from this simple model are also presented in Table 2. The predicted indentation modulus for this bimodal microstructure was calculated to be 114 GPa. This represents a 3.6% difference with respect to the measured modulus of 118 GPa. The predicted indentation yield strength was 1.92 GPa corresponding to a 2.1% difference from the experimentally measured bulk indentation yield strength of 1.96 GPa.

Prior applications of the ROM on multiphase materials have largely employed Equation (8) directly [13,15,16,20–23]. They implicitly assumed that the averaged values of the properties measured from multiple randomly placed indentations in each phase represented adequately the effective property for the respective phases. While this is likely to be true if one employs a very large number of indentations, in practice, this is not the case. In this study, we clearly noticed that Equation (9) produced a significantly better prediction for the bimodal microstructures, compared to the direct use of Equation (8). This is since the large area EBSD scan (see Figure 3) provides much more statistically reliable measure of the different texture components present in the sample. Since the different texture components (with the different declination angles) exhibit very different indentation properties (see Figure 5), Equation (9) represents a much more accurate application of ROM. Since the indentation properties of the basket-weave component did not exhibit a strong dependence on its texture components, there was no need to apply the same approach on the basket-weave component.

It is emphasized again that the consistent use of the spherical indentation stress-strain protocols at both the constituents' scale and the macroscale played an important role in providing a reliable set of measurements for our study. We believe that the protocols described in this work have opened several new research avenues for the critical evaluation and refinement of homogenization models for a broad range of heterogeneous (composite) materials. Furthermore, the high-throughput capabilities of the techniques described and the requirements of relatively small material volumes make these protocols extremely attractive for materials development efforts. This study opens new research avenues into high-throughput multi-resolution studies of the mechanical response of composite materials with complex microstructures.

## 4. Conclusions

This work demonstrates the systematic application of the multi-resolution spherical indentation and microstructure characterization and analyses protocols for generating the data required for an improved understanding of the mechanical response of a complex dual phase metallic alloy sample exhibiting a bimodal microstructure. The sample selected for this study not only comprised of two different crystal structures (i.e., thermodynamic stable phases), but also two different morphologies that exhibited distinct anisotropy in their local mechanical responses. Specifically, the indentation stress-strain curves extracted at the microscale quantified the degree of anisotropy in each constituent. For the primary $\alpha$ phase, it was found that the crystal lattice orientation played an important role in the anisotropic local mechanical response. At the macroscale, the larger micro-indentations were found to consistently and reliably capture the bulk or effective mechanical response of the sample. EBSD analysis allowed the correlation of the local mechanical responses in the individual primary $\alpha$ grains to their crystal lattice orientation. BSE-SEM images were able

to provide reliable measures of the volume fractions of the two main constituents in the bimodal microstructures. All of the information from the microstructural analysis as well as the mechanical characterization was used to evaluate the commonly used Rule of Mixtures (ROM) models for the effective indentation modulus and indentation yield strength of the bimodal microstructures. It was found that the ROM predicted indentation modulus and indentation yield strength were within 4% of the experimentally measured properties.

**Author Contributions:** Conceptualization, N.M.-E. and S.R.K.; methodology, N.M.-E.; formal analysis, N.M.-E.; data curation, N.M.-E.; writing—original draft preparation, N.M.-E. and S.R.K.; supervision, S.R.K.; funding acquisition, S.R.K. All authors have read and agreed to the published version of the manuscript.

**Funding:** The authors acknowledge support from Air Force Office of Scientific Research (AFOSR) Grant FA9550-18-1-0330.

**Conflicts of Interest:** The authors declare no conflict of interest.

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
