# Peer review of "Study of a Bimodal α–β Ti Alloy Microstructure Using Multi-Resolution Spherical Indentation Stress-Strain Protocols"

_jcs, doi:10.3390/jcs6060162_

Round 1
Reviewer 1 Report
The manuscript can be accepted for publication after minor revision.
1. In the last paragraph of page 3, and the first line of page 4, the label of chosen material should be corrected into "Ti6242" instead of Ti6262.
2. The unit of "GPa" was used for the indentation modulus and the indentation yield strength in the manuscript. Thus, it is suggested the unit and the data of Yind should be corrected into the same unit of GPa as the Eind in Table 2.
3. Please check the averaged effective indentation yield strength is 1.99 GPa or 1.96 GPa (1958 ± 101 GPa) according to Table 2 in the manuscript (page 12).
Author Response
Response to Reviewer 1 Comments
Point 1: In the last paragraph of page 3, and the first line of page 4, the label of chosen material should be corrected into "Ti6242" instead of Ti6262.
Response 1: Thank you. The typo has been corrected.
Point 2: The unit of "GPa" was used for the indentation modulus and the indentation yield strength in the manuscript. Thus, it is suggested the unit and the data of Yind should be corrected into the same unit of GPa as the Eind in Table 2.
Response 2: The suggested change has been made in Table 2.
Point 3: Please check the averaged effective indentation yield strength is 1.99 GPa or 1.96 GPa (1958 ± 101 GPa) according to Table 2 in the manuscript (page 12).
Response 3: The value of the averaged effective indentation yield strength has been corrected to be 1.96 GPa (1958 MPa) everywhere in the document and table.
Other comment: English language and style are fine/minor spell check required.
Response: We appreciate your suggestion. The document has been carefully reviewed one more time to correct for these minor failures.

Reviewer 2 Report
Firstly, the paper is interesting for readers as a guide or a protocol that could be used for other researchers interested in this measerement of in the field in general.
The aim of manuscript is not formulated concisely in the last paragraph of Introduction. The authors need to formulate the aim of the work within one or two sentences. Last paragraph could be replaced from Inrtoduction section to Materials and Methods section.
The main results should be presented in Conlusions section without further discussion.
Author Response
Response to Reviewer 2 Comments
Point 1: The aim of manuscript is not formulated concisely in the last paragraph of Introduction. The authors need to formulate the aim of the work within one or two sentences. Last paragraph could be replaced from Introduction section to Materials and Methods section.
Response 1: The suggested change has been made. The last paragraph if the introduction section has been modified to restrict it to the main aim of the study. The other details have been moved to the Materials and Methods section.
Point 2: The main results should be presented in Conclusions section without further discussion.
Response 2: Conclusion section has been edited to keep only the main conclusions of the study.

Reviewer 3 Report
The article is very interesting, it presents a certain way of analyzing the microstructure of titanium materials, which are constantly present in research, and the interest in them is not waning. The article is based on a rich database of world literature.
I noticed some minor problems:
In the introduction, there are quotations of a large number of articles after one sentence, eg [11-26], which can be broken down into smaller groups or the most important literature items can be selected.
I was wondering if the introduction was too extensive, but after reading the entire article, I find it appropriate for this publication.
You should also pay attention to the description of the devices, which does not comply with the recommendations of the journal.
The research is presented in a clear and transparent manner, making it easier to read and analyze later.
I read the article with interest (on Sunday)
best regards
reviewer
Author Response
Response to Reviewer 3 Comments
Point 1: In the introduction, there are quotations of a large number of articles after one sentence, eg [11-26], which can be broken down into smaller groups or the most important literature items can be selected.
Response 1: Citations have been reviewed and only the relevant references have been selected.
Point 2: You should also pay attention to the description of the devices, which does not comply with the recommendations of the journal.
Response 2: Thank you. A more detailed description for the microstructure characterization equipment has been incorporated in the document.
